# MACRPO: Multi-Agent Cooperative Recurrent Policy Optimization

## Abstract

This work considers the problem of learning cooperative policies in multi-agent settings with partially observable and non-stationary environments without a communication channel. We focus on improving information sharing between agents and propose a new multi-agent actor-critic method called *Multi-Agent Cooperative Recurrent Proximal Policy Optimization* (MACRPO). We propose two novel ways of integrating information across agents and time in MACRPO: First, we use a recurrent layer in critic's network architecture and propose a new framework to use the proposed meta-trajectory to train the recurrent layer. This allows the network to learn the cooperation and dynamics of interactions between agents, and also handle partial observability. Second, we propose a new advantage function that incorporates other agents' rewards and value functions by controlling the level of cooperation between agents using a parameter. The use of this control parameter is suitable for environments in which the agents are unable to fully cooperate with each other. We evaluate our algorithm on three challenging multi-agent environments with continuous and discrete action spaces, Deepdrive-Zero, Multi-Walker, and Particle environment. We compare the results with several ablations and state-of-the-art multi-agent algorithms such as MAGIC, IC3Net, CommNet, GA-Comm, QMIX, MADDPG, and RMAPPO, and also single-agent methods with shared parameters between agents such as IMPALA and APEX. The results show superior performance against other algorithms. The code is available online at `https://github.com/kargarisaac/macrpo`.

## 1 Introduction

While reinforcement learning (RL) (Kaelbling et al., 1996) has gained popularity in policy learning, many problems which require coordination and interaction between multiple agents cannot be formulated as single-agent reinforcement learning. Examples of such scenarios include self-driving cars (Shalev-Shwartz et al., 2016), autonomous intersection management (Dresner & Stone, 2008), multiplayer games (Berner et al., 2019; Vinyals et al., 2019), and distributed logistics (Ying & Dayong, 2005). Solving these kind of problems using single-agent RL is problematic, because the interaction between agents and the non-stationary nature of the environment due to multiple learning agents can not be considered (Hernandez-Leal et al., 2019; Lazaridis et al., 2020). Multi-agent reinforcement learning (MARL) and cooperative learning between several interacting agents can be beneficial in such domains and has been extensively studied (Nguyen et al., 2020; Hernandez-Leal et al., 2019).

However, when several agents are interacting with each other in an environment without real-time communication, the lack of communication deteriorates policy learning. In order to alleviate this problem, we propose to share information during training to learn a policy that implicitly considers other agents' intentions to interact with them in a cooperative manner. For example, in applications like autonomous driving and in an intersection, knowing about other cars' intentions can improve the performance, safety, and collaboration between agents.

A standard paradigm for multi-agent planning is to use the centralized training and decentralized execution (CTDE) approach (Kraemer & Banerjee, 2016; Foerster et al., 2016; Lowe et al., 2017; Foerster et al., 2018; Xiao et al., 2021), also taken in this work.

In this work, we propose a new cooperative multi-agent reinforcement learning algorithm, which is an extension to Proximal Policy Optimization (PPO), called *Multi-Agent Cooperative Recurrent Proximal Policy Optimization*

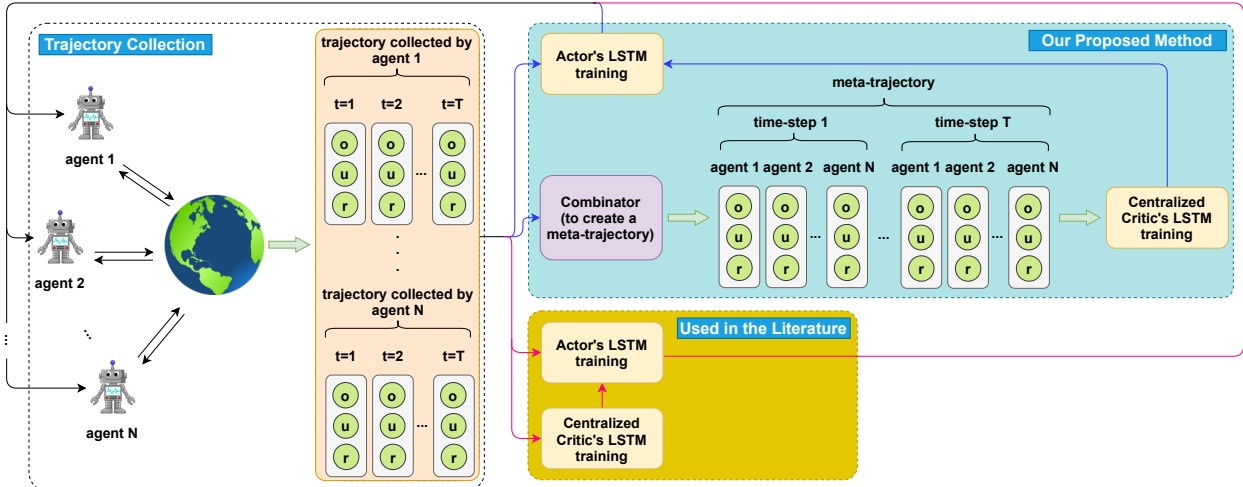

Figure 1: Different frameworks for information sharing. Our proposed method and the standard approach for information sharing through agents are shown in separate boxes. Blue arrows are for ours, and the red ones are for the standard approach to share parameters. After collecting trajectories by agents, ours, in addition to sharing parameters between agents, uses the meta-trajectory to train the critic's LSTM layer. This allows the critic to learn the interaction between agents along the trajectories through its hidden state. In contrast, the literature approach, which does parameter sharing, uses separate trajectories collected by agents to train the LSTM layer. For more details about the network architectures, please see Fig 2.

(MACRPO). MACRPO combines and shares information across multiple agents in two ways: First, in network architecture using long short term memory (LSTM) layer and train it by creating a meta-trajectory from trajectories collected by agents, as shown in Fig 1. This allows the critic to learn the cooperation and dynamics of interactions between agents, and also handle the partial observability. Second, in the advantage function estimator by considering other agents' rewards and value functions.

MACRPO uses a centralized training and decentralized execution paradigm that the centralized critic network uses extra information in the training phase and switches between agents sequentially to predict the value of a state for each agent. In the execution time, only the actor networks are used, and each learned policy (actor network) only uses its local information (i.e., its observation) and acts in a decentralized manner.

Moreover, in environments with multiple agents that are learning simultaneously during training, each agent's policy and the dynamics of the environment, from each agent's perspective, is constantly changing. This causes the non-stationarity problem (Hernandez-Leal et al., 2019; Xiao et al., 2021). To reduce this effect, MACRPO uses an on-policy approach and the most recent collected data from the environment.

In summary, our contributions are as follows: (1) proposing a cooperative on-policy centralized training and decentralized execution framework that is applicable for both discrete and continuous action spaces; (2) sharing information across agents using two ways: a recurrent component in the network architecture which uses a combination of trajectories collected by all agents and an advantage function estimator that uses a weighted combination of rewards and value functions of individual agents which uses a control parameter that can be utilized to change the cooperation level between agents in MARL problems; (3) evaluating the method on three cooperative multi-agent tasks: DeepDrive-Zero (Quiter, 2020), Multi-Walker (Gupta et al., 2017), and Particle (Mordatch & Abbeel, 2018) environments, demonstrating similar or superior performance compared to the state-of-the-art.

The rest of this paper is organized as follows. The review of related works in Section 2 demonstrates that while MARL has been extensively studied, existing approaches do not address the dynamics of interaction between agents in detail. In Section 3, we provide the required background in Markov Games and Proximal Policy Optimization. The problem definition and the proposed method are described in Section 4, with emphasis on the two innovations, meta-trajectory for recurrent network training and joint advantage function. Then, Section 5 presents empirical evaluation in three multi-agent environments showing superior performance of the proposed approach compared to state-of-the-art.

Finally, in Section 6 we conclude that implicit information sharing can be used to improve cooperation between agents while discussing its limitations in settings with high number of agents.

## 2 RELATED WORK

The most straightforward and maybe the most popular approach to solve multi-agent tasks is to use single-agent RL and consider several independent learning agents. Some prior works compared the performance of cooperative agents to independent agents, and tried independent Q-learning (Tan, 1993) and PPO with LSTM layer (Bansal et al., 2017), but they did not work well in practice (Matignon et al., 2012). Also, Zhao et al. (2020) tried to learn a joint value function for two agents and used PPO with LSTM layer to improve the performance in multi-agent setting.

In order to use single-agent RL methods for multi-agent setting, improve the performance, and speed up the learning procedure, some works used parameter sharing between agents (Gupta et al., 2017; Terry et al., 2020b). Especially in self-play games, it is common to use the current or older versions of the policy for other agents (Berner et al., 2019). We will compare our proposed method with several state-of-the-art single-agent RL approaches with shared parameters between agents proposed in Terry et al. (2020b) in the experiments section. Our way of training the LSTM layer in the critic differs from parameter sharing used in the literature such that instead of using separate LSTMs for each agent, the LSTM layer in our method has a shared hidden state, which is updated using a combination of all agents' information. This lets the LSTM layer to learn about the dynamics of interaction and cooperation between agents and across time.

In addition to using single-agent RL methods with or without parameter sharing, some other works focused on designing multi-agent RL algorithms for multi-agent settings. In multi-agent environments, considering communication between agents and information sharing will lead to designing multi-agent methods (Niu et al., 2021; Singh et al., 2019; Liu et al., 2020; Sukhbaatar et al., 2016; Dutta et al., 2005; Da Silva & Costa, 2019; Kash et al., 2011). The communication channel is often limited, leading to methods that try to optimize the communication including message structure (Mao et al., 2020; Kullu et al., 2017). However, in some environments, there is no explicit communication channel between agents. For example, consider an autonomous driving environment without connection between cars. Finding a solution to address this problem and decrease the lack of communication effect seems necessary.

A recently popularized paradigm to share information between agents is to use centralized training and decentralized execution. In general, we can categorize these types of approaches into two groups: value-based and actor-critic-based. In value-based methods, the idea is to train a centralized value function and then extract the value functions for each agent from that to act in a decentralized manner in the execution time (Sunehag et al., 2018; Rashid et al., 2018). On the other hand, the actor-critic-based methods have actor and critic networks (Lowe et al., 2017; Foerster et al., 2018). The critic network has access to data from all agents and is trained in a centralized way, but the actors have only access to their local information. They can act independently in the execution time. The actors can be independent with individual weights (Lowe et al., 2017) or share the policy with shared weights (Foerster et al., 2018). In this work, we use an actor-critic-based method with centralized training and decentralized execution, providing two innovations to improve information sharing without communication channel between agents during execution.

RMAPPO (Yu et al., 2022) is a method close to ours, which uses CTDE framework. They make no mention of recurrent neural networks (RNNs) in their paper, but their code contains recurrent layers. In addition, they concentrate primarily on adapting PPO components such as clipping, mini-batching, batch size, value normalization, value function input representation, etc. for multi-agent environments. The distinction between our work and theirs is the meta-trajectory we generate from the data of all agents and the specific manner in which we employ the RNN layer, whereas they employ CTDE and RNN as usual without a combined trajectory as input. To share information, they use a shared policy between all agents which is similar to what we do in addition to the meta-trajectory idea. Also, they have a shared reward function for all agents which is the sum of all agents' rewards without any cooperation control parameter. In addition, their implementation and benchmark environments all use discrete action spaces, while we test our method on both discrete and continues action spaces.

In Foerster et al. (2018), which is another work near ours, the actor is recurrent, but the critic is a feed-forward network, whereas our actor and critic are both recurrent, and the recurrent layer in our critic has a crucial role in our method. Their method is also for settings with discrete action spaces, whereas we test our method on three environments with both discrete and continuous action spaces.

ROLA (Xiao et al., 2021) is another work near ours. They use LSTMs in both actor and critic networks. Additionally, ROLA employs both centralized and individual asymmetric critics that estimate individual advantage values using local history and/or state information. However, we construct the meta-trajectory which has not only the history of each agent, but also the history of the interaction between agents and the environment's dynamics. In addition, we propose a novel advantage function estimator which is a combination of all agents' advantage functions and the cooperation level of agents can be changed based on the problem using a control parameter.

Durugkar et al. (2020) is also a work that combines the agent specific reward and an environment-specific reward to accomplish the shared task. They consider a framework that uses a linear mixing scheme to balance individual preferences and task rewards. They demonstrate that in their test environments, a small amount of selfishness and not full cooperation can be advantageous and facilitate team learning. In our test environments and with our framework, full cooperation among agents yields superior performance. Depending on the environment, the amount of cooperation and selfishness can be different.

The other similar work to ours, which is one of the most popular MARL methods, is the multi-agent deep deterministic policy gradient (MADDPG) (Lowe et al., 2017) that proposed similar frameworks with centralized training and decentralized execution. They tested their method on some Particle environments (Mordatch & Abbeel, 2018). Their approach differs from ours in the following ways: (1) They do not have the LSTM (memory) layer in their network, whereas the LSTM layer in the critic network plays a critical role in our method. It helps to learn the interaction and cooperation between agents and also mitigate the partial observability problem. (2) They tested MADDPG on Multi-Agent Particle Environments with discrete action spaces. But we test our method in both continuous and discrete action space environments. (3) They consider separate critic networks for each agent, which is beneficial for competitive scenarios, whereas we use a single critic network and consider the cooperative tasks. (4) Their method is off-policy with replay buffer, and they combat the non-stationarity problem by centralized training. In contrast, our approach, in addition to centralized training, is an on-policy method without replay buffer allowing the networks to use the most recent data from the environment. We will compare our method with MADDPG and show that ours has comparable or superior performance. Wang et al. (2020) extends the MADDPG idea and adds a recurrent layer into the networks, but they have separate actors and critics for agents, similar to MADDPG, and recurrent hidden states of critics are isolated, and there is no combination of information in them. They also tested their method on one environment with a discrete action space.

We target problems where agents attempt to collaboratively maximize the sum of all agents' expected rewards but where each agents receives its own reward. We do not specifically consider the credit assignment problem for multi-agent games where all agents have a shared team reward. The proposed algorithm can be applied to such problems, but it is not designed for them.

## 3 BACKGROUND

### 3.1 MARKOV GAMES

In this work, we consider a multi-agent extension of Partially Observable Markov Decision Processes (MPOMDPs) (Gmytrasiewicz & Doshi, 2005), also called partially observable Markov games (Littman, 1994). It can also be modeled as a partially observable stochastic games (POSGs) (Hansen et al., 2004). A Markov game for $N$ agents is defined by a set of states $\mathcal{S}$ describing the possible configurations of all agents, a set of actions $\mathcal{U}_1, \ldots, \mathcal{U}_N$ and a set of observations $\mathcal{O}_1, \ldots, \mathcal{O}_N$ for each agent. The probability distribution of the next state as a function of current state and actions is determined by a Markovian transition function $\mathcal{T} : \mathcal{S} \times \mathcal{U}_1 \times \ldots \times \mathcal{U}_N \to \mathcal{S}$. Each agent $i$ uses a stochastic policy $\pi_{\theta_i} : \mathcal{O}_i \times \mathcal{U}_i \to [0, 1]$, parametrized by $\theta_i$, to choose an action. Upon the state transition, the agent receives a scalar reward $r_i : \mathcal{S} \times \mathcal{U}_i \to \mathbb{R}$. We consider games where the total reward can be decomposed to individual agent rewards $r_i$. Each agent $i$ aims to maximize the rewards for all agents in a cooperative way (Lowe et al., 2017).

### 3.2 Proximal Policy Optimization

Proximal Policy Optimization (PPO) is a family of policy gradient methods for solving reinforcement learning problems, which alternate between sampling data through interaction with the environment, and optimizing a surrogate

objective function using stochastic gradient descent while limiting the deviation from the policy used to collect the data (Schulman et al., 2017). PPO aims to maximize the clipped expected improvement of the policy

$$L^{CLIP}(\theta) = \hat{\mathbb{E}}_t[min(f_t(\theta)\hat{A}_t, clip(f_t(\theta), 1 - \epsilon, 1 + \epsilon)\hat{A}_t)]$$

where $\hat{A}_t$ is the advantage obtained by Generalized Advantage Estimation (GAE), $\epsilon$ is a hyperparameter, and $f_t(\theta)$ denotes the probability ratio $f_t(\theta) \equiv \frac{\pi_\theta(u_t|o_t)}{\pi_{\theta_{old}}(u_t|o_t)}$ for importance sampling. The clipping prevents excessively large policy updates.

In addition to the expected improvement, the total objective function for PPO incorporates a loss function for a critic network required for GAE and an entropy bonus term to encourage exploration, resulting in the total objective (Schulman et al., 2017)

$$L_t^{CLIP+VF+S}(\theta) = \hat{\mathbb{E}}_t[L_t^{CLIP}(\theta) - c_1 L_t^{VF}(\theta) + c_2 S[\pi_\theta](o_t)] \tag{1}$$

where $c_1$, $c_2$ are weight factors, $S$ denotes the entropy bonus, and $L_t^{VF}$ is a squared-error loss for the critic

$$L_t^{VF}(\theta) = (V_\theta(o_t) - V_t^{targ})^2 \tag{2}$$

In the above equations, $V_\theta(o_t)$ is the state-value function and $\theta$ denotes the combined parameter vector of actor and critic networks. PPO uses multiple epochs of minibatch updates for each set of sampled interactions.

## 4 Method

In this section, we first explain the problem setting and outline our proposed solution. We then proceed to describing its two main components: a critic based on a recurrent neural network with a new proposed meta-trajectory and advantage estimation using weighted rewards with a control parameter for cooperation level between agents, ending with a summary of the proposed algorithm.

### 4.1 Problem Setting and Solution Overview

Information sharing across agents can help to improve the performance and speed up learning (Gupta et al., 2017; Foerster et al., 2018; Terry et al., 2020b). In this work, we focus on improving information sharing between agents in multi-agent settings in addition to just sharing parameters across actors. We propose Multi-Agent Cooperative Recurrent Proximal Policy Optimization (MACRPO) algorithm, which is a multi-agent cooperative algorithm and uses the centralized learning and decentralized execution framework. In order to improve information sharing between agents, MACRPO, in addition to parameter sharing, uses two novel ideas: (a) a recurrent critic architecture that is trained using a meta-trajectory created by combining trajectories collected by all agents (Section 4.2), and (b) an advantage function estimator that combines the rewards and value functions of individual agents using a control parameter that can be employed to alter the degree of cooperation amongst agents. (Section 4.3).

### 4.2 MACRPO Framework

The proposed MACRPO framework consists of one recurrent actor, similar to Foerster et al. (2018), and one recurrent critic network, as illustrated in Fig. 1. To consider the partial observability of multi-agent settings, we use recurrent LSTM layers in both actor and critic networks to allow integration of information over time.

The actor network architecture is composed of a stack of Embedding, LSTM, and Linear layers and is trained using trajectories collected by all agents. We denote the shared weights of actors with $\theta_a$ and use the same, latest weights for all agents. The behaviors of different agents vary because of stochasticity and difference in their inputs. Denoting the trajectory data for episode $k$ with length $T$ for agent $i$ as

$$\tau_i^k = (o_1^i, u_1^i, r_1^i, \ldots, o_T^i, u_T^i, r_T^i),$$

the training data for the actor is then $D_A = (\tau_1^1, \ldots, \tau_i^k, \ldots)$.

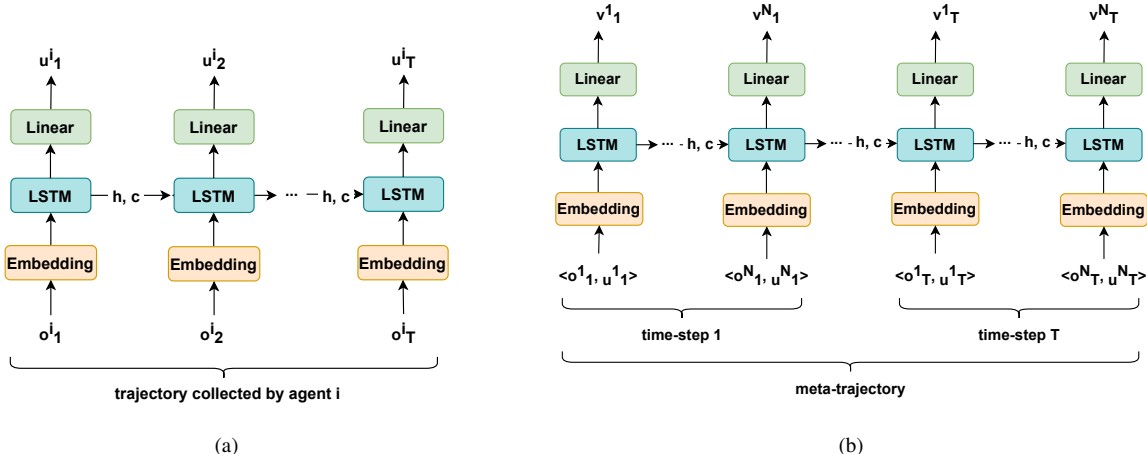

Figure 2: Actor and critic network architectures. (a) Actor network architecture for agent $i$ which uses the collected trajectory by itself, (b) The centralized critic network architecture which uses the created meta-trajectory. Note that $u$, $v$, and $o$ denote action, value, and observation, respectively. Also, superscripts and subscripts show agent number and time-step, respectively.

To allow the critic network, which is also a stack of Embedding, LSTM, and Linear layers, to integrate information across agents and time, we use all agents' trajectories in each roll-out and concatenate them in a sequence to create a meta-trajectory, and train the critic network using that (see Fig. 1). To remove the dependency to the order of agents, we randomize the order of agents at each meta-trajectory generation phase. Whenever we create a meta-trajectory, we consider one order and create the meta-trajectory and train the network with it. So the order stays the same in that meta-trajectory. We then change the order of agents for creation of the next meta-trajectory, but it remains the same for the entire meta-trajectory. Similar to the training data for actor, we can define the training data for the critic network too. Denoting the meta-trajectory for episode $k$ with length $T$ for $N$ agents as

$$\mu^k = (o_1^1, \ldots, o_1^N, u_1^1, \ldots, u_1^N, r_1^1, \ldots, r_1^N, \ldots$$
$$, o_T^1, \ldots, o_T^N, u_T^1, \ldots, u_T^N, r_T^1, \ldots, r_T^N)$$

the training data for the critic is then $D_C = (\mu^1, \ldots, \mu^k, \ldots)$.

By using the above meta-trajectory, the critic network receives information from all agents to capture the agents' history, the interactions between them, and the environment dynamics, all capture by the hidden state. In other words, MACRPO is able to consider temporal dynamics using the LSTM layer, which incorporates a history of states and actions across all agents. Modeling temporal dynamics allows the latent space to model differential quantities such as the rate of change (derivative) between the distance of two agents and integral quantities such as the running average of the distance.

Additionally, the hidden state of recurrent networks can be viewed as a communication channel that allows information to flow between agents to create richer training signals for actors during training. The network will update the hidden state in each time-step by getting the previous hidden state and the data from the agent $i$ in that time-step. The network architectures for actor and critic are shown in Fig 2. It is important to note that the critic network is only needed during training and that the optimized policy can be deployed using only the actor such that the agents are able to operate in a fully distributed manner without communication.

## 4.3 Objective Function

In addition to the LSTM layer, we propose a novel advantage function estimator based on weighted discounted returns using a parameter which controls the agents' cooperation level and integrates information across agents. We consider the $V_t^{targ}$ in Equation (2) as discounted return and propose to calculate it for agent $i$ at time $t$ as

$$R_t^i = \bar{r}_t + \gamma \bar{r}_{t+1} + \ldots + \gamma^{T-t+1} \overline{V}(o_T^i) \tag{3}$$

---

**Algorithm 1** MACRPO

1: Randomly initialize actor and critic networks' parameters $\theta_c$ and $\theta_a$
2: **for** iteration=1, 2, ... **do**
3:     **for** environment=1, 2, ..., E **do**
4:         Run all N agents with latest trained weights in the environment for T time-steps and collect data
5:         Combine collected trajectories by all agents according to Fig 1
6:         Compute discounted returns and advantage estimates using Equations (5, 3)
7:     **end for**
8:     **for** epoch=1, ..., K **do**
9:         **for** minibatch=1, ..., M **do**
10:           Calculate the loss functions using Equations (7, 8)
11:           Update Actor and Critic parameters via Adam
12:         **end for**
13:     **end for**
14: **end for**

---

where

$$\overline{r}_t = \frac{r_t^i + \beta \sum_{j \neq i} r_t^j}{N}, \quad \overline{V}(o_T^i) = \frac{V(o_T^i) + \beta \sum_{j \neq i} V(o_T^j)}{N} \tag{4}$$

where $r_t^i$ is the reward for agent $i$ at time $t$, $\gamma$ is the discount factor, $\beta$ is the cooperation control parameter used for rewards of other agents, and $V(o_T^i)$ is the value for the final state of agent $i$. The advantage for each agent $i$ is then calculated as

$$\hat{A}_t^i = \delta_t^i + (\gamma\lambda)\delta_{t+1}^i + \ldots + \ldots + (\gamma\lambda)^{T-t+1}\delta_{T-1}^i \tag{5}$$

where

$$\delta_t^i = \frac{1}{N}[r_t^i + \gamma V(o_{t+1}^i) - V(o_t^i) + \\ + \beta \sum_{j \neq i}(r_t^j + \gamma V(o_{t+1}^j) - V(o_t^j))] \tag{6}$$

where $\lambda$ is the temporal difference factor of the GAE algorithm, and $V(o_t^i)$ is the state-value at time $t$ for agent $i$.

The intuition behind the weighting is that each agents' own rewards are likely to be affected most by its own action choice but that the actions taken by other agents can also affect the reward. In addition, the $\beta$ parameter can be interpreted as a control parameter for cooperation level between agents. This heuristic is related to credit assignment between agents and provides a trade-off between optimizing the policy considering only individual rewards ($\beta = 0$ and no cooperation between agents), which could lead to sub-optimal total reward when individual rewards are in conflict with each other, and optimizing the policy using the sum of all rewards ($\beta = 1$ and full cooperation between agents), which could lead to challenging assignment of credit between agents. One should note that policy optimization is performed across all agents such that in the end, the expected rewards over all agents are maximized, independent of the choice of $\beta$.

MACRPO uses separate networks for actor and critic. Therefore, the objective functions of the actor and critic networks are separate, in contrast to PPO. The actor's objective function in the shared weights case is defined as

$$L_t^{CLIP+S}(\theta_a) = \hat{\mathbb{E}}_t[L_t^{CLIP}(\theta_a) + cS[\pi_{\theta_a}](o_t)] \tag{7}$$

and the critic objective function is

$$L_t^{VF}(\theta) = (V_{\theta_c}(o_t) - V_t^{targ})^2 \tag{8}$$

where $\theta_c$ are the parameters of the critic A parallelized version of the MACRPO algorithm is shown in Algorithm 1.

## 5 EXPERIMENTS

This section presents empirical results that compare the performance of our proposed method, MACRPO, with several ablations to see the effect of each proposed novelty. We also compare our method with recent advanced RL meth-

ods in both single-agent domain with shared parameters between agents (Gupta et al., 2017; Terry et al., 2020b) and multi-agent domain like MAGIC, (Niu et al., 2021), IC3Net (Singh et al., 2019), GA-Comm (Liu et al., 2020), Comm-Net (Sukhbaatar et al., 2016), MADDPG (Lowe et al., 2017), RMAPPO (Yu et al., 2022), and QMIX (Rashid et al., 2018).

## 5.1 Test Environments

We test our method in three MARL environments. In two of them, DeepDrive-Zero (Quiter, 2020) and Multi-Walker (Terry et al., 2020a) environments, the action space is continuous, and in the third environment, the Particle environment (Mordatch & Abbeel, 2018), the action space is discrete. Fig 3 show these three environments.

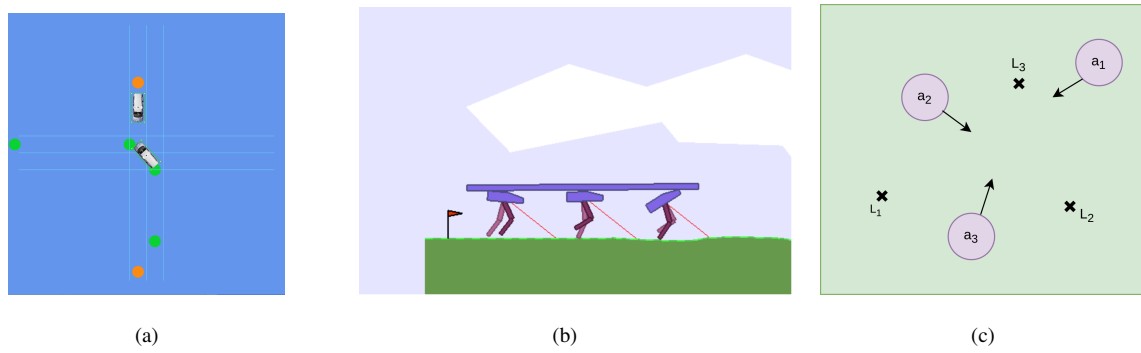

|     (a)     |     (b)     |     (c)     |

Figure 3: Considered MARL simulation environments (a) DeepDrive-Zero environment: an unprotected left turn scenario, (b) Multi-Walker environment, (c) Particle environment: cooperative navigation.

**DeepDrive-Zero Environment:** There are several autonomous driving simulators which can be used for multi-agent simulation (Dosovitskiy et al., 2017; Santara et al., 2021; Quiter, 2020). In this work, we use DeepDrive-Zero (Quiter, 2020), because we don't need to deal with image data and also need a fast simulation environment for training. DeepDrive-Zero is a very fast and 2D simulation environment for self-driving cars which uses a bike model for the cars. We use the unsignalized intersection scenario in this work, which is shown in Fig 3a. To test our algorithm, we consider two cars in the environment, one starts from the south and wants to follow the green waypoints to do an unprotected left-turn, and the other one starts from the north and wants to go to the south and follow the orange waypoints. The agents need to learn to cooperate and negotiate to reach their destination without any collision.

**Multi-Walker Environment:** The multi-walker environment is a multi-agent continuous control locomotion task introduced in Gupta et al. (2017). The environment contains agents (bipedal walkers) that can actuate the joints in each of their legs and convey objects on top of them. Fig 3b shows a snapshot from the environment.

**Cooperative Navigation in Particle Environment:** Using the particle environment package from OpenAI (Lowe et al., 2017), we created a new environment based on the cooperative navigation environment. This new environment consists of N agents and N landmarks, and agents must avoid collisions and cooperate to reach and cover all landmarks. Fig 3c shows the simulation environment.

Check Appendix A for more details about the environments.

## 5.2 Ablation Study

Four ablations were designed to evaluate each novelty. The name of the method and the explanation shows which ablation has Feed-forward or LSTM or how information is shared in that ablation. In all cases, the parameter sharing proposed in Gupta et al. (2017) and Terry et al. (2020b) was used:

**FF-NIC** *(Feed-forward multi-layer perceptron (MLP) network + no information combination)*: two feed-forward neural networks for actor and critic. The GAE is calculated using the single-agent PPO GAE equation (Schulman et al., 2017). There is no LSTM layer or reward and value functions combination for information sharing in this case.

**FF-ICA** *(Feed-forward MLP network + information combination using the advantage estimation function)*: This case is similar to the previous case, but the GAE is calculated using Equation (5) to show the effect of mixing reward and value functions for information sharing. There is no LSTM layer in this case too.

**LSTM-NIC** *(LSTM network + no information combination)*: two networks with LSTM layers for actor and critic. There is no information sharing between agents through GAE calculation or the LSTM's hidden state. The GAE is calculated using the single-agent PPO GAE equation (Schulman et al., 2017).

**LSTM-ICA** *(LSTM network + information combination using the advantage estimation function but not through the LSTM layer)*: This case is identical to the previous case, but the GAE is calculated using Equation (5).

**LSTM-ICF** *(LSTM network + information sharing using both the advantage estimation function and an LSTM layer in the critic network (full proposed method))*: two networks with LSTM layers for actor and critic. In addition to parameter sharing between actors, the information integration is done through both the advantage estimation function and the LSTM's hidden state in the centralized critic network, shown in Fig 1.

Also, in order to see the effect of the $\beta$ value in Equations (4, 6), the proposed method was evaluated with different $\beta$ values which shows different cooperation levels between agents.

All experiments were repeated with identical random seeds for each method to reduce the effect of randomness. Hyperparameters used in MACRPO for three environments are detailed in Appendix C.

**DeepDrive-Zero Environment:** We ran all ablations for ten random seeds in the DeepDrive-Zero environment to test our proposed method. We used self-play in simulations and used the latest set of parameters for actors in each episode. The results are shown in Fig 4a. The x-axis shows the number of training iterations. In each iteration, we ran 100 parallel environments for 3000 steps and collected data. Next, we updated actors and critic networks using the collected data. After each iteration, we ran the agents for 100 episodes, took the mean of these episodes' rewards (sum of all agents' rewards), and plotted them. The shaded area shows one standard deviation of episode rewards. The hyperparameters used in the MACRPO algorithm are listed in Table 2 in Appendix C.

The proposed algorithm, LSTM-ICF, outperforms the ablations. The next best performances are for LSTM-ICA and FF-ICA, which are almost the same. Moreover, information integration in the advantage function, in both FF-ICA and LSTM-ICA, improves the performance compared to FF-NIC and LSTM-NIC; however, the achieved performance gain in the fully connected case is higher. The FF-ICA surpasses LSTM-NIC, which shows the effectiveness of sharing information across agents through the proposed advantage function, even without an LSTM layer. Furthermore, the addition of LSTM layer to add another level of information integration, LSTM-ICF, boosts performance when compared to FF-ICA. Fig 4b shows the analysis of the effect of different $\beta$ values in Equations (3, 4, 6). The best performance is for $\beta = 1$, which is for the full cooperation between agents, and as the value of $\beta$, agents' cooperation level, is reduced, the agents' performance decreases. We demonstrate the effect of different $\beta$ values in this environment, but for other environments, the results will be provided for $\beta \in \{0, 1\}$ only.

To achieve smooth driving performance, a curriculum-based learning method and gradual weight increase of reward factors were used. The weights of Jerk, G-force, steering angle change, acceleration change, and going out of the lane in the reward function were gradually increased to $3.3 \times 10^{-6}$, 0.1, 3, 0.05, and 0.3, respectively. We then added termination of episodes for lane violation to force cars to stay between the lanes. After curriculum learning and smoothing the driving behavior, the cars follow the waypoints to reach their destination. The car that starts from the bottom and wants to make a left-turn yields nicely for the other agent if they reach the intersection simultaneously and then make the left-turn, and if it has time to cross the intersection before the other agent arrives, it does. A video of the final result can be found in the supplementary materials.

**Multi-Walker Environment:** We ran 20 parallel environments and 2500 time-steps during each update iteration for the Multi-Walker environment. After each iteration, we ran agents for 100 episodes and plotted the mean of these episodes' rewards. Each episode's reward is the sum of all the agents' rewards. Ten different random seeds are used for each ablation. We also used the latest set of parameters for all actors. The hyperparameters used in the MACRPO algorithm are listed in Table 2 in Appendix C.

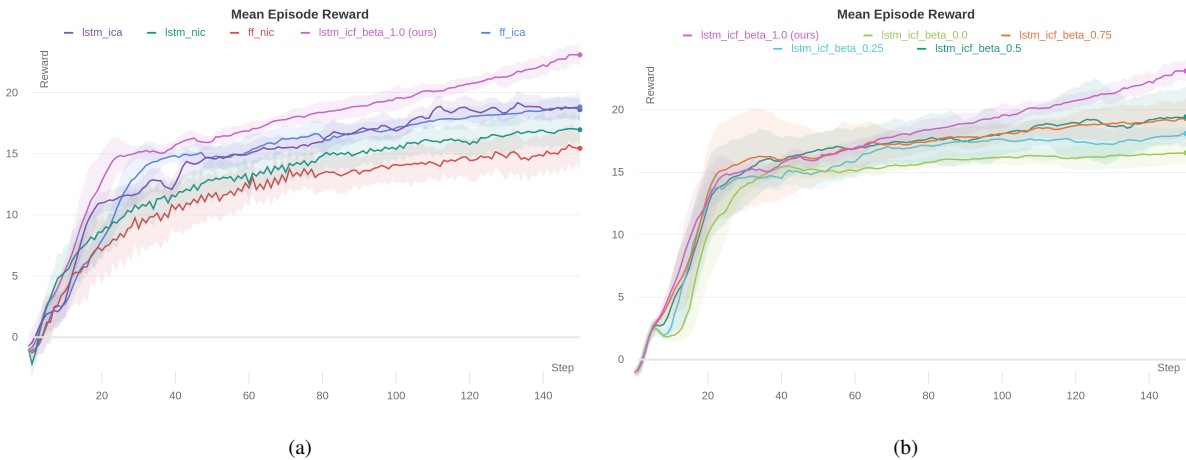

Figure 4: Simulation results in DeepDrive-Zero environment. (a) Mean episode reward for different ablations, (b) mean episode reward for different $\beta$ values. The shaded area shows one standard deviation.

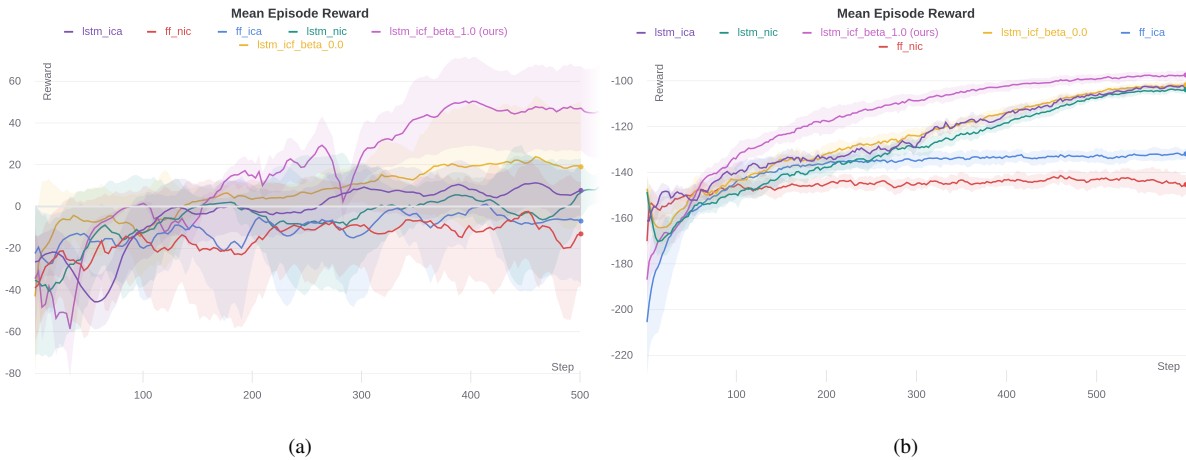

Figure 5: Simulation results in Multi-Walker and Particle environments for different ablations. (a) Multi-Walker simulation results, (b) Particle environment simulation results.

Fig 5a shows a massive performance improvement of our proposed method, LSTM-ICF with $\beta = 1$, when compared to ablations. LSTM-ICF with $\beta = 0$, information integration through only the LSTM layer, has the next best performance. After these two, LSTM-ICA, which does the information integration using the advantage estimation function, performs better than FF-ICA, FF-NIC, and LSTM-NIC cases. The effect of $\beta$ value and information sharing through the advantage estimation function in performance improvement can be seen as we move from LSTM-ICF with $\beta = 0$ to LSTM-ICF with $\beta = 1$ and from FF-NIC to FF-ICA. By comparing FF-ICA and LSTM-ICF, we can also see the impact of information integration using the LSTM layer. Note that the $\beta$ value in FF-ICA is equal to 1. A video of the trained model can be found in the supplementary materials.

**Cooperative Navigation in Particle Environment:** In the particle environment, in each iteration, we ran 20 parallel environments to collect data for 2500 time steps and used that data to update the network. The agents were then evaluated using the trained weights for 100 episodes. We ran the simulation with six random seeds. MACRPO hyperparameters are shown in Table 2 in Appendix C.

The results of this environment are depicted in Fig 5b. Similar to the other two environments, the proposed LSTM-ICF with $\beta = 1$ outperforms ablations. The next best performance is achieved with LSTM-ICF with $\beta = 0$, which only uses the LSTM layer that was trained using the created meta-trajectory. Moreover, the LSTM-ICA's performance

Table 1: Comparing performance of our method with state-of-the-art approaches. Numbers show average reward in each environment for ten random seeds, except for the Multi-Walker environment which is 1000 random seeds.

| Method | DeepDrive-Zero | Multi-Walker | Particle |
|---|---|---|---|
| DQN | 4 | -100000 | -151.8 |
| RDQN | 6 | -100000 | 153.2 |
| A2C | 0.5 | -27.6 | -148.6 |
| DDPG | 2 | -57.8 | - |
| PPO | 16 | 41 | -144.3 |
| SAC | -1.5 | -16.9 | -143.7 |
| TD3 | -1 | -8 | - |
| APEX-DQN | 8 | -100000 | -136.2 |
| APEX-DDPG | 14 | -23 | - |
| IMPALA | -0.66 | -88 | -155.2 |
| MADDPG | -0.1 | -96 | -98.3 |
| QMIX | -0.9 | -24 | -155.6 |
| MAGIC | 3.1 | - | -114 |
| IC3Net | 2.1 | - | -117 |
| GA-Comm | 1.9 | - | -119 |
| CommNet | 1.6 | - | -115 |
| RMAPPO | -0.43 | - | -131 |
| Ours ($\beta = 0$) | 17.3 | 24.2 | -100.7 |
| Ours (full model) | **23.7** | **47.8** | **-95.8** |

is almost identical to LSTM-ICF when $\beta = 0$. This shows that both novel ideas cause the same performance gain over LSTM-NIC. These results show that cases with LSTM layer perform better than feed-forward ones, even in the FF-ICA case, which integrates information through the advantage function. A video of the trained model can be found in the supplementary materials.

Both ideas were evaluated in the ablation study, and the results clearly demonstrate the effect of the proposed ideas in performance improvement. Ablation studies provide evidence that the findings are not spurious, but are associated with the proposed enhancements. $\beta = 1$ corresponds to the total reward over all agents, the optimization goal. However, it is known that such a team reward causes a credit assignment problem since each agent's contribution to the team reward could differ. Due to this, we wanted to experimentally study whether beta values less than one would alleviate the credit assignment problem to the extent that the suboptimality of the reward would be overcome. According to the results of the experiment, this wasn't the case, and $\beta = 1$ gave the best performance.

Moreover, as the results illustrate, both proposed ideas result in a performance gain, but this is not the same for all environments. In the DeepDrive-Zero environment, information integration through advantage function estimation improves the performance slightly more than the LSTM layer. However, in the Multi-Walker environment, the LSTM layer is more effective, and in the Particle environment, their effect is almost the same.

## 5.3 Comparison to State-of-the-Art Methods

We compared the proposed method with several state-of-the-art algorithms in each environment. Our method is compared against several single-agent baselines with shared parameters across agents (DQN, RDQN, A2C, DDPG, PPO, SAC, TD3, APEX-DQN, APEX-DDPG, and IMPALA), which were tested in Terry et al. (2020b). We also compared our method to state-of-the-art multi-agent approaches such as MAGIC, (Niu et al., 2021), IC3Net (Singh et al., 2019), GA-Comm (Liu et al., 2020), CommNet (Sukhbaatar et al., 2016), MADDPG (Lowe et al., 2017), RMAPPO (Yu et al., 2022), and QMIX (Rashid et al., 2018).

The architecture and hyperparameters used for PPO, IMPALA, A2C, SAC, APEX-DQN, Rainbow-DQN, DQN, APEX-DDPG, DDPG, TD3, and QMIX are taken from Terry et al. (2020b) which has an open source implementation [1]. For MADDPG (Lowe et al., 2017), we used the original source code [2], for RMAPPO (Yu et al., 2022) we used their open source source code [3], and for MAGIC, IC3Net, CommNet, and GA-Comm we used the open source implementation [4]. We performed hyperparameter tuning using grid search to optimize performance for each method.

Note that some of the official implementations of baselines we used here do not support both discrete and continues action spaces and we did not modify the code. The non-reported results for some baselines in the paper's tables and charts are due to this. In addition, we tried to use the discritized version of the DeepDrive-Zero environment for algorithms with discrete action space which may cause poor performance.

Each agent's goal in MACRPO is to maximize the total reward of all agents, while the goal of other methods is to maximize the total reward of each agent without considering other agents' reward in their objective function. In order to have a more fair comparison, We report the result for our method when $\beta = 0$ too. The results are shown in Table 1. The table contains some empty fields due to the fact that some algorithms do not support continuous or discrete action spaces. Check Appendix B for more details.

**DeepDrive-Zero Environment:** In this environment, our full method and also the case with $\beta = 0$ achieved the highest average reward. The next best was PPO with parameter sharing between agents followed by APEX-DQN and APEX-DDPG. A version of the environment with discretized action space was used for algorithms with discretized action space.

**Multi-Walker Environment:** Similar to the previous environment, the proposed method outperformed other methods by a large margin with an average reward of 47.8. Next, PPO with parameter sharing had the second-best performance with a maximum average reward of 41. Our method with $\beta = 0$ achieved the third best average reward. Some algorithms do not support continuous action spaces and are marked with a dash in the table.

**Cooperative Navigation in Particle Environment:** As in both previous environments, our approach outperformed other approaches in this environment as well, although the difference was minor compared to MADDPG. Our method with $\beta = 0$ is in the third place after MADDPG with small margin. We used a categorical distribution instead of a multivariate Gaussian distribution in this environment with discrete action space. Algorithms with continuous action spaces were not tested in this environment, and are marked with a dash in the table. Adapting these algorithms for discrete action environments could be achieved using the same trick, but we did not change the standard implementation for baselines.

It is evident from the reported results that RMAPPO performance in the DeepDrive-Zero environment is not satisfactory and that it is average in the Particle environment. As the current implementation of RMAPPO does not support continuous action spaces, we could not test this method in the Multi-Walker environment. Additionally, we conducted limited hyperparameter searches for RMAPPO, but since this method aims to recommend a set of modifications and hyperparameters that will improve PPO's performance for multi-agent systems, we did not deviate too far from the main hyperparameters. The performance of MADDPG is not also good in DeepDerive-Zero and Multi-Walker environments. However, it performs well when used in the Particle environment.

All hyperparameters for each algorithm are included in Appendix C.

The results show that the performance benefit given by the two proposed ways of sharing information across agents is significant such that the method outperforms state-of-the-art algorithms.

---

[1]https://github.com/parametersharingmadrl/parametersharingmadrl
[2]https://github.com/openai/maddpg
[3]https://github.com/marlbenchmark/on-policy
[4]https://github.com/CORE-Robotics-Lab/MAGIC

# 6 Conclusion & Future Work

In this paper, MACRPO, a centralized training and decentralized execution framework for multi-agent cooperative settings was presented. The framework is applicable to both discrete and continuous action spaces. In addition to parameter sharing across agents, this framework integrates information across agents and time in two novel ways: network architecture and the advantage estimation function. An ablation study in three environments revealed that both ways of information sharing are beneficial. Furthermore, the method was compared to state-of-the-art multi-agent algorithms such as MAGIC, IC3Net, CommNet, GA-Comm, QMIX, and MADDPG, as well as single-agent algorithms that share parameters between agents, such as IMPALA and APEX. The results showed that the proposed algorithm performed significantly better than state-of-the-art algorithms. A single recurrent network to summarize the state of all agents may be problematic when the number of agents is large. A potential solution to this problem could be to use an attention mechanism for the agent to learn on which other agents to pay attention to, warranting further study to realize the potential of the proposed approach with a high number of agents.

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

## A  Detailed Environment Descriptions

**DeepDrive-Zero Environment:**  The observation space is a vector with continuous values. Each agent in the environment receives some information about itself, as well as information from other agents. This information can come from some modules like Perception, Localization, and HDMap in a self-driving car and be used by the decision making and control modules. The observation vector for each agent contains some information about the agent itself like distance and angle to waypoints, velocity, acceleration, and distance to left and right lanes, and also some information about the other agents like the relative velocity of the other agent to the ego agent, velocity and acceleration of the other car, angles to corners of the other agent, and distance to corners of the other agent.

Each action vector element is continuous from -1 to 1: steering, acceleration, and braking. Negative acceleration can be used to reverse the car, and the network outputs are scaled to reflect physically realistic values. This environment also has a discretized version that we used in discrete action methods.

The reward function is a weighted sum of several terms like speed, reaching the destination, collision, G-force, jerk, steering angle change, acceleration change, and staying in the lane. Initially, we used 0.5, 1, 4, $1 \times 10^{-7}$, $6 \times 10^{-6}$, 0.0001, 0.0001, 0.001 as weights, then used curriculum learning to smooth the driving behavior.

**Multi-Walker Environment:** To keep the package balanced and move it as far to the right as possible, the walkers must coordinate their movements. A positive reward is given to each walker locally, based on the change in the package distance summed with 130 times the change in the walker's position. A walker is given a reward of -100 if they fall, and all walkers receive a reward of -100 if the package falls while moving forward has a reward of 1. By default, the environment is done whenever a walker or package falls or when the walkers reach the edge of the terrain. The action space is continuous, with four values for torques applied to each walker's leg. The observation vector for each walker is a 32-dimensional vector that contains information about nearby walkers as well as data from some noisy LiDAR sensors.

**Cooperative Navigation in Particle Environment:** We assign each agent a landmark and calculate its local reward based on its proximity to its landmark and collisions with other agents. As a result, agents will have different reward values; not one shared reward. Each agent's observation data is its position and velocity, as well as the relative position of other agents and landmarks. There are five discrete actions in the action space: up, down, left, right, and no move. After 25 time-steps, the episode ends.

## B   Comparison to State of The Art Methods

To get a better idea of the performance of the state-of-the-art algorithms, the mean episode reward for different baseline algorithms in test environments are shown in Fig 6.

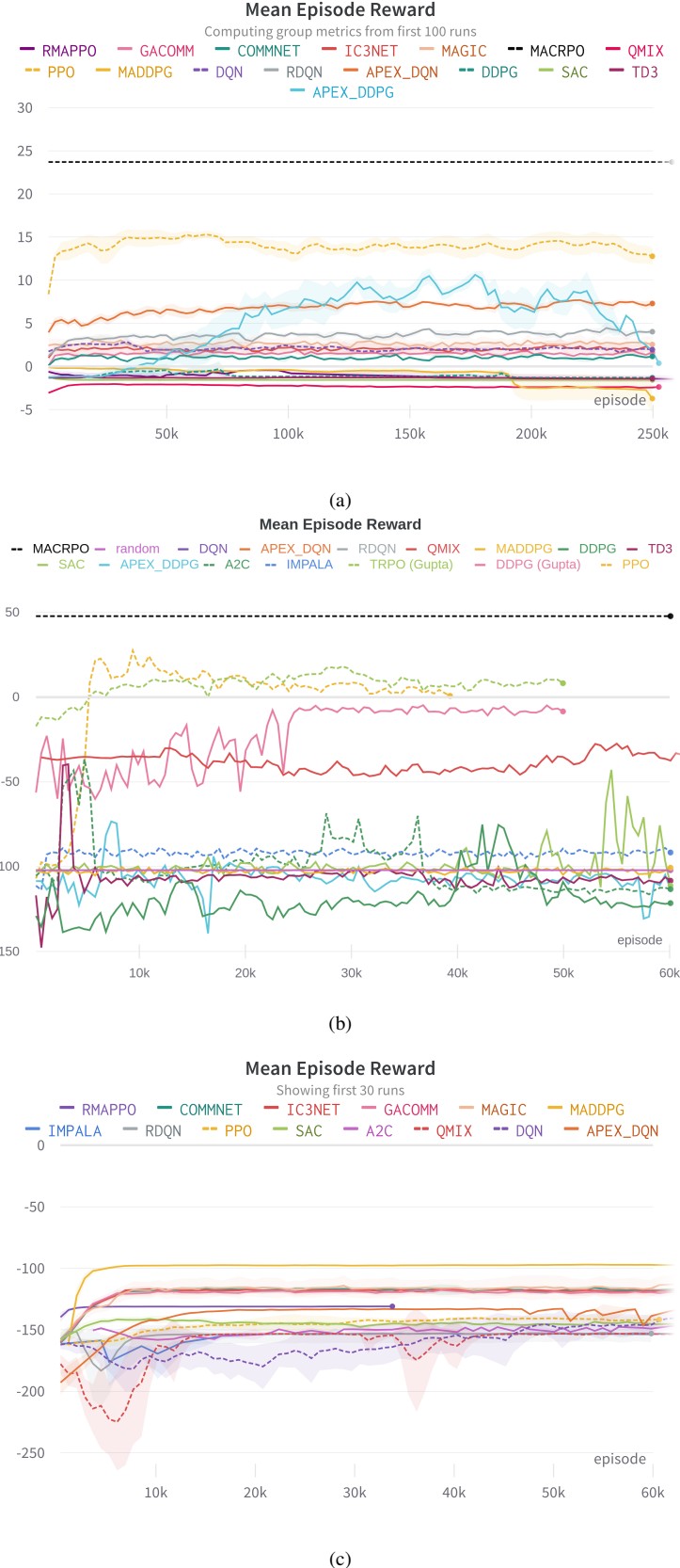

Figure 6: Analysis of baseline algorithms proposed in Terry et al. (2020b) in three environments: (a) DeepDrive-Zero, (b) Multi-Walker, and (c) Particle environments.

## C Hyperparameters

Hyperparameters used in MACRPO for three environments are described in Table 2.

Table 2: MACRPO hyperparameters for three MARL environments

| Param. | DeepDrive-Zero | Multi-Walker | Particle |
|---|---|---|---|
| actor hidden size | 64 | 32 | 128 |
| critic hidden size | 128 | 32 | 128 |
| batch size | 512 | 32 | 1500 |
| discount | 0.99 | 0.99 | 0.99 |
| GAE lambda | 0.94 | 0.95 | 0.95 |
| PPO clip | 0.15 | 0.3 | 0.2 |
| PPO epochs | 4 | 4 | 10 |
| max grad norm | 1.0 | 1.0 | 1.0 |
| entropy factor | 0.001 | 0.01 | 0.01 |
| learning rate | 0.0002 | 0.001 | 0.005 |
| recurrent sequence length (time-step) | 20 | 40 | 3 |
| no. of recurrent layers | 1 | 1 | 1 |

The architecture and hyperparameters used for other baselines are taken from Terry et al. (2020b) with some fine-tuning to get better performance, and are shown in Tables 3, 4, and 5. Some hyperparameter values are constant across all RL methods for all environments. These constant values are reported in Table 6. We used the source code for all algorithms from Terry et al. (2020b) except for MADDPG which we used the original implementation (Lowe et al., 2017).

Table 3: Hyperparameters for three MARL environments

| RL method | Hyperparameter | DeepDrive-Zero | Multi-Walker | Particle |
|---|---|---|---|---|
| PPO | sample_batch_size | 100 | 100 | 100 |
| | train_batch_size | 5000 | 5000 | 5000 |
| | sgd_minibatch_size | 500 | 500 | 1000 |
| | lambda | 0.95 | 0.95 | 0.95 |
| | kl_coeff | 0.5 | 0.5 | 0.5 |
| | entropy_coeff | 0.01 | 0.01 | 0.001 |
| | num_sgd_iter | 10 | 10 | 50 |
| | vf_clip_param | 10.0 | 10.0 | 1.0 |
| | clip_param | 0.1 | 0.1 | 0.5 |
| | vf_share_layers | True | True | True |
| | clip_rewards | True | True | False |
| | batch_mode | truncate_episodes | truncate_episodes | truncate_episodes |
| IMPALA | sample_batch_size | 20 | 20 | 20 |
| | train_batch_size | 512 | 512 | 512 |
| | lr_schedule | [[0, 5e-3], [2e7, 1e-12]] | [[0, 5e-3], [2e7, 1e-12]] | [[0, 5e-3], [2e7, 1e-12]] |
| | clip_rewards | True | True | False |
| A2C | sample_batch_size | 20 | 20 | 20 |
| | train_batch_size | 512 | 512 | 512 |
| | lr_schedule | [[0, 7e-3], [2e7, 1e-12]] | [[0, 7e-3], [2e7, 1e-12]] | [[0, 7e-3], [2e7, 1e-12]] |
| SAC | sample_batch_size | 20 | 20 | 20 |
| | train_batch_size | 512 | 512 | 512 |
| | Q_model | {activation: relu, layer_sizes: [266, 256]} | {activation: relu, layer_sizes: [266, 256]} | {activation: relu, layer_sizes: [266, 256]} |
| | optimization | {actor_lr: 0.0003, actor_lr: 0.0003, entropy_lr: 0.0003,} | {actor_lr: 0.0003, actor_lr: 0.0003, entropy_lr: 0.0003,} | {actor_lr: 0.0003, actor_lr: 0.0003, entropy_lr: 0.0003,} |
| | clip_actions | False | False | False |
| | exploration_enabled | True | True | True |
| | no_done_at_end | True | True | True |
| | normalize_actions | False | False | False |
| | prioritized_replay | False | False | False |
| | soft_horizon | False | False | False |
| | target_entropy | auto | auto | auto |
| | tau | 0.005 | 0.005 | 0.005 |
| | n_step | 1 | 1 | 5 |
| | evaluation_ interval | 1 | 1 | 1 |
| | metrics_smoothing_ episodes | 5 | 5 | 5 |
| | target_network_ update_freq | 1 | 1 | 1 |
| | learning_starts | 1000 | 1000 | 1000 |
| | timesteps_per_ iteration | 1000 | 1000 | 1000 |
| | buffer_size | 100000 | 100000 | 100000 |

Table 4: Hyperparameters for DeepDrive-Zero, Multi-Walker, and Particle environments

| RL method | Hyperparameter | DeepDrive-Zero | Multi-Walker | Particle |
|---|---|---|---|---|
| APEX-DQN | sample_batch_size | 20 | 20 | 20 |
| | train_batch_size | 32 | 512 | 5000 |
| | learning_starts | 1000 | 1000 | 1000 |
| | buffer_size | 100000 | 100000 | 100000 |
| | dueling | True | True | True |
| | double_q | True | True | True |
| Rainbow-DQN | sample_batch_size | 20 | 20 | 20 |
| | train_batch_size | 32 | 512 | 1000 |
| | learning_starts | 1000 | 1000 | 1000 |
| | buffer_size | 100000 | 100000 | 100000 |
| | n_step | 2 | 2 | 2 |
| | num_atoms | 51 | 51 | 51 |
| | v_min | 0 | 0 | 0 |
| | v_max | 1500 | 1500 | 1500 |
| | prioritized_replay | True | True | True |
| | dueling | True | True | True |
| | double_q | True | True | True |
| | parameter_noise | True | True | True |
| | batch_mode | complete_episodes | complete_episodes | complete_episodes |
| Plain DQN | sample_batch_size | 20 | 20 | 20 |
| | train_batch_size | 32 | 512 | 5000 |
| | learning_starts | 1000 | 1000 | 1000 |
| | buffer_size | 100000 | 100000 | 100000 |
| | dueling | False | False | False |
| | double_q | False | False | False |
| QMIX | buffer_size | 10000 | 3000 | 100000 |
| | gamma | 0.99 | 0.99 | 0.99 |
| | critic_lr | 0.001 | 0.0005 | 0.001 |
| | lr | 0.001 | 0.0005 | 0.001 |
| | grad_norm_clip | 10 | 10 | 10 |
| | optim_alpha | 0.99 | 0.99 | 0.99 |
| | optim_eps | 0.00001 | 0.05 | 0.00001 |
| | epsilon_finish | 0.02 | 0.05 | 0.02 |
| | epsilon_start | 1.0 | 1.0 | 1.0 |
| MADDPG | lr | 0.001 | 0.0001 | 0.01 |
| | batch_size | 64 | 512 | 500 |
| | num_envs | 1 | 64 | 1 |
| | num_cpus | 1 | 8 | 1 |
| | buffer_size | 1e5 | 1e5 | 1e5 |
| | steps_per_update | 4 | 4 | 4 |

Table 5: Hyperparameters for DeepDrive-Zero and Multi-Walker

| RL method | Hyperparameter | DeepDrive-Zero | Multi-Walker |
|---|---|---|---|
| APEX-DDPG | sample_batch_size | 20 | 20 |
| | train_batch_size | 512 | 512 |
| | lr | 0.0001 | 0.0001 |
| | beta_annealing_fraction | 1.0 | 1.0 |
| | exploration_fraction | 0.1 | 0.1 |
| | final_prioritized_replay_beta | 1.0 | 1.0 |
| | n_step | 3 | 3 |
| | prioritized_replay_alpha | 0.5 | 0.5 |
| | learning_starts | 1000 | 1000 |
| | buffer_size | 100000 | 100000 |
| | target_network_update_freq | 50000 | 50000 |
| | timesteps_per_iteration | 2500 | 25000 |
| Plain DDPG | sample_batch_size | 20 | 20 |
| | train_batch_size | 512 | 512 |
| | learning_starts | 5000 | 5000 |
| | buffer_size | 100000 | 100000 |
| | critics_hidden | [256, 256] | [256, 256] |
| TD3 | sample_batch_size | 20 | 20 |
| | train_batch_size | 512 | 512 |
| | critics_hidden | [256, 256] | [256, 256] |
| | learning_starts | 5000 | 5000 |
| | pure_exploration_steps | 5000 | 5000 |
| | buffer_size | 100000 | 100000 |

Table 6: Variables set to constant values across all RL methods for all environments

| Variable | Value set in all RL methods |
|---|---|
| # worker threads | 8 |
| # envs per worker | 8 |
| gamma | 0.99 |
| MLP hidden layers | [400, 300] |

