# OpenReview forum: "MACRPO: Multi-Agent Cooperative Recurrent Policy Optimization"
_TMLR — Rejected by TMLR_

### Review · Reviewer_V5DY · 2022-10-19

**Summary Of Contributions:**

This paper deals with cooperative multi-agent RL where each agent is given its own reward signal by the environment.
In order to encourage cooperation in such a setting, one prevalent approach that avoids an explicit communication channel is the paradigm of centralized training and decentralized execution (CTDE).

This paper considers a CTDE actor critic setting. In particular, it uses a PPO based actor critic setup where the critic in centralized and the actors are decentralized.

This paper claims to propose two innovations to this setup:
1. Use of an LSTM in the critic
2. Scaling and adding other agents' rewards to each agent in order to encourage coordination.

It tests these two modifications in 3 domains, and compares to some previous CTDE algorithms, as well as some independent RL algorithms.

**Audience:**

Yes

**Broader Impact Concerns:**

Effective multi-agent RL training could help scale the deployment of intelligent agents that learn while interacting with the environment.
This paper essentially considers a problem of training agents to maximize social good.
The necessity that agents with super-human policies do not act solely in their own interest is of wide societal interest. These points are not discussed at all in the manuscript. Some discussion along these lines, with perhaps some reference to works that try to incorporate social good in the training mechanisms, would be good to have.

**Claims And Evidence:**

Yes

**Requested Changes:**

* Response to questions above.
* Clarification as to how the proposed contributions differ from the approaches in [1] and [2] from section above. If these approaches are found to be related, a comparison with them in the experiments should also be included.
* Updating related work section to more accurately depict MADDPG.
* In Background, the paper introduces partially observable Markov games as MPOMDPs. While this is an acceptable way to denote that each agent is in a POMDP from its own perspective, a more elegant framework to communicate the setting would be as partially observable stochastic games (POSGs) [3]. This change is up to the discretion of the authors.
* In the ablations, an ablation with a centralized critic but no RNN.
* Short descriptions of the environments should be in the main document. In the appendix, more details about each environment would be more helpful, such as the exact dimensionality of the state and action spaces, the number of agents trained, etc.

### References:
[3] Hansen, E.A., Bernstein, D.S. and Zilberstein, S., 2004, July. Dynamic programming for partially observable stochastic games. In AAAI (Vol. 4, pp. 709-715).

**Strengths And Weaknesses:**

### Strengths:
* The explanation of the proposed techniques is clear for the most part.
* Comparison to related work are done well for the most part.
* Code has been shared openly (though without anonymization)
* The experimental evaluation on 3 domains with various baselines is well done.
* The ablations testing the proposed components are, for the most part, evaluating these additions appropriately.

### Weaknesses and Questions:
* I am not convinced that either of the proposed innovations have not been tried before. Details below.
* R-MAPPO [1] already uses PPO in a CTDE setting with an RNN in the critic. It also combines the individual agent information for a more informed critic. At the very least, this work should be compared to, and should be explained and contrasted with appropriately in related work.
* The idea of combining individual agent rewards with the shared team rewards is also not unique. Previous work has balanced individual objectives and shared rewards. Look at [2] for example.
* On page 5, the paper states: `To remove the dependency to the order of agents, we randomize the order by permuting all inputs to the LSTM layer according to a random order of agents at each time step`. This permutation of the order at each time step and its utility is unclear as a reader. Won't this make the problem of encouraging coordination unnecessarily hard for the centralized critic? If ordering is changed at each time step the critic won't be able to keep track of which observations are which agents. Is there some nuance that has been overlooked?
* The experiments seem to indicate that $\beta=1$, meaning that the agents are learning with a fully shared team reward, is the one that does best. This result also seems to bear out across domains. Given this result, should one of the insights of this paper be that teams of agents should be trained with shared rewards if cooperation is paramount? How would this result contrast with that from [2]?
* Could the mixed reward scheme be tested on some of the other baselines that are compared with, to quantify the benefit of this reward sharing mechanism?
* In related work, the paper claims that MADDPG and follow up work is tested on environments with discrete action spaces. This cannot be true since DDPG is a continuous action space specific algorithm.

### References:
[1] Yu, C., Velu, A., Vinitsky, E., Wang, Y., Bayen, A. and Wu, Y., 2021. The surprising effectiveness of ppo in cooperative, multi-agent games. arXiv preprint arXiv:2103.01955.
[2] Durugkar, I., Liebman, E. and Stone, P., 2021, January. Balancing individual preferences and shared objectives in multiagent reinforcement learning. In Proceedings of the Twenty-Ninth International Conference on International Joint Conferences on Artificial Intelligence (pp. 2505-2511).

---

### Review · Reviewer_duKs · 2022-10-21

**Summary Of Contributions:**

This paper proposes a version of MAPPO that uses a recurrent network for the critic. They also add a term that adds the rewards of the other agents to a particular agent's reward, inducing cooperation. These changes are very incremental. Experiments perform equally well or outperform baselines, but do not compare to MAPPO.

**Audience:**

No

**Claims And Evidence:**

Yes

**Requested Changes:**

Add experiments comparing to MAPPO.

Acknowledgements should be anonymized.

**Strengths And Weaknesses:**

The main strength of this paper is the experimental results, where the method performs better than some baselines. However, notably absent is MAPPO (https://arxiv.org/abs/2103.01955) which is not even cited. So since the method is a very small modification of MAPPO but isn't compared to MAPPO, I do not think members of the community will find this work interesting as is. Perhaps with strong comparisons against MAPPO, and principled reasons why the method performs better than MAPPO then people would find it interesting.

---

### Review · Reviewer_J9Ez · 2022-12-14

**Summary Of Contributions:**

The authors are studying Cooperative multi-agent reinforcement learning (but with defined individual rewards)  and propose an algorithm combining existing components in a new manner and perform an experimental comparison across three domains where they show their algorithm having the best performance.  A strength is that it works on both continuous and discrete domains.

A key point in their architecture is to feed in a combined trajectory from all actors into a centralized critic to enable it to better learn about the interactions. I am aware of marl works with individual rewards where individually embedding each trajectory and combing has been utilized for purposes of scaling to many agents, dealing with variable number of agents and permutation invariance within groups but I am not sure to what extent centralized critics for coop-marl has utilized this technique. The original VDN underlying Q-mix had comparisons with agents embedding and combining each trajectory into one LSTM and then outputting individual heads in a DQN style approach, which also differs from the use in the paper.

Another utilized technique is making the individuals other-regarding (common topic/technique in sequential social dilemmas) and having the reward (and values) used to learn from, incorporating a term that is the average for the other agents. In the case of team rewards, this term would not make a difference as they all have the same reward.




**Audience:**

Yes

**Claims And Evidence:**

Yes

**Requested Changes:**

Add a SCII comparison.
Look into finding coop-marl baselines that outperforms PPO or explain why a simple single-agent algorithm is the second best algorithm across the domains chosen for a evaluating coop-marl approaches.
If there is no reason why the same method used to have MADDPG evaluated on discrete domain cannot be applied for DDPG and APEX-DDPG, then please complete the table in this manner.

**Strengths And Weaknesses:**


As the introduced algorithm is incremental as in combining existing ideas the experimental evaluation is very important.

The experimental comparison is performed on three domains. They show that they outperform the state-of-the-art though one immediately notices the state-of-the-art losing to a simple single-agent algorithm like PPO which is at first confusing. Q-mix, usually a strong performer in coop-marl, is terrible in this comparison. A reason for this is probably that they have applied it to action discretizatized versions of continuous domains, which might lead to very bad performance. They have two continuous domains and one discrete. They say that they do not show results for the continuous actions methods in the discrete domain, though for MADDPG they do apply it with a categorical distribution instead  of the gaussian used in continuous domains, but does not do the same for DDPG and APEX-DDPG which I am not sure if there is a good reason for. Further, as much recent coop-marl work has been performed on the SCII micromanagement suite of tasks based on which the VDN/QMIX/… class of algorithms has continued its development it would have been good to see how the new proposed approach compared there where the notion of QMIX being state-of-the-art comes from.

I find the comparison is a bit unconvincing, as the chosen state-of-the-art coop-marl is performing much worse than generic single-agent algorithms on the chosen domains as they have been applied. Is there some particular weakness of all existing coop-marl approaches that are highlighted by these domains and the new approach addresses it? Is there no stronger coop-marl approach for these domains, or application/setup of the chosen approaches.

Besides the comparison to other approaches, the authors are proposing a nice ablation study of their design choices.

---

### Decision · Action_Editors · 2023-02-03

**Recommendation:** Reject

**Comment:**

As outlined in Claims and Evidence, reviewers were not sufficiently convinced by the experimental evaluation, and thus unanimously agreed that the paper is not yet ready for publication. The paper would be strengthened by bolstering reader confidence in the experimental results, in particular in explaining why (via intuition building or further experiments) previously SOTA MARL methods seem to perform worse than single-agent methods. Along these lines, reviewers suggest further hyperparameter tuning of baselines and comparison on the SCII suite of micromanagement tasks.

**Audience:**

I believe that, due to reviewer lack of confidence in the current experimental evaluation, there is a limited audience for the work as is.

**Claims And Evidence:**

Reviewers had concerns about novelty, given that the proposed method seems to mostly combine existing methods. This then places more of a burden on the experimental evaluation, but unfortunately, there too, the reviews had concerns. Reviewers were surprised to see previously SOTA methods perform poorly (more poorly than single agent methods) without a convincing explanation of why this is (and worried that it might be due to a lack of hyperparameter tuning). Reviewers suggested running a comparison on the SCII suite of tasks, where there is more existing work to compare to, but the authors felt there was not time during the discussion to do this. In summary, reviewers felt that the current work lacks sufficient experimental support for the superiority of the proposed method.